



# Technical note: Pleistocene climate sensitivity to $CO_2$ forcing is path dependent in reconstructions

Roger M. Cooke[1], Willy P. Aspinall[2]

[1]Resources for the Future, Washington DC, 200036 USA
[2] School of Earth Sciences, University of Bristol, Bristol, UK

*Correspondence to: Roger M. Cooke (cooke@rff.org)*

**Abstract.** Improved high-resolution paleo records of atmospheric carbon dioxide ($CO_2$) concentrations and reconstructions of Earth's surface temperature are available. We analyse one authoritative Pleistocene dataset to explore how the climate
sensitivity parameter $S$ varies under different system states, using linear regression of mean annual surface temperature changes against $CO_2$ forcing changes. Data are partitioned by *path* (deglaciation or glaciation). On the whole data set, $S = 2.04K/Wm^{-2}$ and $CO_2$ forcing explains *64%* of the variance in temperature. During deglaciation periods, $S = 2.34K/Wm^{-2}$, explaining *75%* of the temperature variance.; during glaciations, $S = 1.59K/Wm^{-2}$ and explains *48%* of the temperature variance. Possible process-related explanations for these *path*-related differences are conjectured.

# 1 Introduction

The climate sensitivity parameter $S$ is somewhat loosely defined in the literature (Myhre et al 2013) as the change with respect to present in mean annual global surface temperature ($\Delta T$) per unit change in forcing with respect to present ($\Delta F$). $S$ is sometimes estimated from data by linear regression: $\Delta T = S \times \Delta F + B + e$ with intercept $B$ and error $e$. $\Delta F$ is $[Wm^{-2}]$ so that $S$ is $[K/(Wm^{-2})]$. The variance of $e$ is estimated as $(SS - SSR)/(N - df - 1)$ where $N$ is the number of observations, $df$ is
the number of parameters in the model (here always =1), $SS$ is the sum square deviations from the population mean and $SSR$ is the sum square deviations of the regression estimates from the population mean. $SSR/SS$ is the fraction of variance of $\Delta T$ explained by the regression model, termed $R^2$. The lower is $R^2$ the more of the variation in $\Delta T$ must be attributed to factors other than $\Delta FCO_2$.

It is widely believed that "… the change in surface temperature is directly proportional to the radiative forcing. Hence, this becomes the simplest way of quantifying the effect in a perturbation in greenhouse gas inventory" (Byrne and Goldblatt 2014). Note that $\Delta T$ can be proportional to $\Delta F$ only if the intercept $B$ is zero. Assuming $B = 0$ forces the trend line to pass through the origin ($\Delta F = 0$, $\Delta T = 0$). This can inflate and skew the errors and can cause the regression model to be a worse predictor of the data than simply predicting the mean of $\Delta T$ for each value of $\Delta F$ (as happens here, see Figure 1 and



associated discussion). Others argue that the regression line "needs to pass through the origin to avoid any biases" (Köhler et
al 2017). Different statistical packages have different interpretations of $R^2$ when $B = 0$ is stipulated, but none has the
interpretation as fraction of explained variance (see SI), thereby disabling this important diagnostic. In what follows, we
refrain from stating $R^2$ values if $B$ is set to zero. Of course, the placement of the origin effects the value of $S$ if $B = 0$ is
stipulated. For example, Snyder (2019) considers $\Delta T$ relative to the average temperature over the last five thousand years,
whereas Martinez-Boti et al (2015) define temperature change relative to pre-industrial temperature. If the intercept is
estimated this choice is immaterial. In any event, the consequences of stipulating $B = 0$ are large and should be carefully
weighed.

Here, we utilize the dataset (Martinez-Boti et al 2015) for the Pleistocene (*1096* records from *0.14* to *798.51 kaBP* (thousand
years before present). A second Pleistocene dataset (Snyder, 2019) yields very similar results and is discussed briefly. In
recent years, several researchers have studied paleoclimate data for evidence of "state dependence" in the climate sensitivity
parameter (e.g. Meraner et al 2013) It has been observed that the linear approximation breaks down in the long tail of high
climate sensitivity commonly seen in observational studies (Bloch-Johnson et al 2015). Transient behavior of climate
sensitivity has been explored using an energy balance model combined with observational and modeling CMIP5 constraints
(Goodwin, 2018). Background state dependence and tipping points in Earth system sensitivity with millennial timescales
(von der Heydt et al 2016) motivate the introduction of the more general "climate sensitivity parameter". Others find that
climate sensitivity strongly depends on the climate background state, with significantly larger values attained during warm
phases (Meraner et al 2013. Friedrich et al 2016). Some authors voice concerns over the simple concepts underlying climate
sensitivity and radiative forcing (Knutti et al 2010). Non-linear dependence of land ice albedo forcing is found (Köhler et al
2010), while non-linearity of $CO_2$ forcing is said to depend on the $CO_2$ data set. Global mean temperature and $CO_2$ diverge
during intervals of pronounced land ice growth (Köhler et al 2018). The need to distinguish actuo- and paleoclimate
sensitivity over different time scales is emphasized (Rohling et al 2018). Averaged glacial and interglacial climate
sensitivities are estimated (Shao et al 2019) using Earth system model simulations of the Last Glacial Cycle.

The present study applies standard regression analysis to study variations in the climate sensitivity parameter, using
Pleistocene data from Martínez-Botí et al (2015). Rather than estimating non-linear regression functions, we partition the
data into periods of increasing versus periods of decreasing paths of $CO_2$ concentration corresponding to deglaciation versus
glaciation respectively. The SI also looks at partitions into epochs before and after 424 *kaBP* and periods of low,
intermediate and high $CO_2$ concentration. The path dependence is the most important followed by epochal dependence.
Dependence on background $CO_2$ concentration has low explanatory power but does interact with the other two (see SI).

To compute the fraction of explained variance it is necessary to estimate the intercept term. Comparison with regressions in
which the intercept $B$ is set to zero shows that the latter have higher values of the climate sensitivity parameter with much



less variation over the partition elements. While this data may be subject to errors, random errors would depress the $R^2$

values. The wide variation of $R^2$ values across the partition elements argues against a strong random or systematic error

across the entire data set.

Given the wide variations in $S$, projections of regression equations to estimate $CO_2$ concentrations out of sample cannot be

considered predictions unless the conditions going forward are very similar to the conditions on which such regressions are

based.

## 2. Analysis of Pleistocene data

A deterministic functional relation between changes in $CO_2$ concentrations and changes in the induced forcing $(\Delta FCO_2)$ is

inferred from radiative transfer codes, where $278\ ppmv$ is taken as the pre-industrial concentration of $CO_2$:

$$\Delta FCO_2\ (CO_2) = 5.3515 \times ln(CO_2/278)\ [Wm^{-2}]. \tag{1}$$

Note that the forcing change depends only on the ratio of two $CO_2$ concentrations and not on their actual values. This

assumption is pervasive in the literature. Doubling $CO_2$ with respect to pre-industrial:

$$\Delta FCO_2\ (556) = 3.71\ [Wm^{-2}]. \tag{2}$$

Regressing mean annual surface temperature $(\Delta MAT$ in Martinez-Boti et al 2015) on $\Delta FCO_2$ gives the following equation

which explains 64% of the variance in $\Delta T$:

$$\Delta T(\Delta FCO_2) = 2.037 \times \Delta FCO_2 - 0.709 + e \tag{3}$$

The climate sensitivity parameter $S = 2.037[K/Wm^{-2}]$. The values of $S$ and $B$ in eqn. *(3)* are chosen to minimize the squared

distance between the values of $\Delta T$ and the linear trend.

To project the effect on temperature of doubling $CO_2$ above pre-industrial, based on the whole Pleistocene data, substitute

$\Delta FCO_2 = 3.71$ in (3) and find $\Delta T(556) = +6.85°K$. Neglecting the intercept term would result in: $2.037[K/Wm^{-2}] \times 3.71\ [W]$



$= +7.56\,\mathrm{\text{°}K}$. This, of course, is incorrect: if we wish to constrain the intercept to be zero, we must find the line passing
through the origin minimizing squared distance to $\Delta T$. In that case, $S = 2.531 K/Wm^{-2}$, $\Delta T(556) = +9.391\,\mathrm{\text{°}K}$ (Figure 1).

Eqn. (1) is just a positive affine transformation of $ln(CO_2)$. If we regress $\Delta T$ on $ln(CO_2)$, we would also explain 64% of the
variance and also find $\Delta T(556) = +6.85\,\mathrm{\text{°}K}$. The only difference would be that the units of the linear coefficient would be
$[K/ln(ppmv)]$, instead of $[K/Wm^{-2}]$. The latter dimension suggests physical agency; indeed, a *GHG* induced radiative
imbalance at the top of the atmosphere causes warming according to the Stefan Boltzmann law. At the same time, a rise in
temperature can raise atmospheric $CO_2$ concentrations through temperature mediated feedbacks as, for example, when higher
ocean temperatures reduce $CO_2$ uptake in the oceans. We retain the familiar dimension of $K/Wm^{-2}$ for the climate sensitivity
parameter while recognizing that it reflects a choice of units rather than physical agency.

For the Pleistocene data of Martinez-Boti et al (2015), the mean $\Delta T = -2.816\,\mathrm{\text{°}K}$. Letting $i$ index the *1096* data points,
summing the squared deviations from the mean of the black regression line ($B = 0$) in Figure 1 left gives $SSR = \Sigma_i\ (2.531\times$
$\Delta FCO_2(i) +2.816)^2 = 2969 > 2942 = \Sigma_i\ (\Delta T(i) + 2.816)^2 = SS$. Using the black line as a predictor of $\Delta T$ is a bit worse than
predicting the population mean of $\Delta T$ for each value of $\Delta FCO_2(i)$. For the red regression line ($B$ estimated), $SSR = \Sigma_i$
$(2.037\times \Delta FCO_2(i) - 0.709 +2.816)^2 = 1896$, which explains *1896/2942 = 0.644 (64%)* of the variance of $\Delta T$.

Neglecting to clarify whether the intercept is inferred or stipulated invites confusion. In this data, setting $B = 0$ inflates the $S$
values and suppresses the differences over different partition elements. Table 1 compares results with and without the
intercept assumption.

**2. Partitioning the data**

Figure 2 shows $CO_2$ as a function of age in *kaBP*, with local minima and maxima identified with blue *resp* red arrows. This
enables us to distinguish episodes of increasing (deglaciation) and decreasing (glaciation) $CO_2$ concentrations. Note that
increasing $CO_2$ episodes generally transpire more quickly than decreasing $CO_2$ episodes. Dividing the data into two subsets
consisting of increasing or decreasing episodes allows us to determine whether the climate sensitivity parameter is $CO_2$ -
path dependent. Figure 2 also suggests that the Earth's climate system changed around the inception of Marine Isotope
Stage 11 in 424 *kaBP*, midway between a glacial maximum and a glacial minimum. The SI examines epochal dependence by
comparing sensitivity in the periods before and after 424 *kaBP* and the results are in Table 1.





The results show evidence of path dependence. Note that if the intercept is estimated the differences in the climate sensitivity

parameter between glaciation and deglaciation *(1.5885, 2.3352)* are larger than if the intercept is stipulated to be zero *(2.5051, 2.552)*. The same is true for the projected temperature at *522ppmv $CO_2$:* with the intercept estimated these are *(4.72, 8.32)* but with intercept stipulated these are *(9.16, 9.47)*. The right panel of Figure 2 showing glaciation also illustrates how stipulating the intercept can lead to inferior predictions. Summing over the 583 data points in the right panel, the square differences between $\Delta T$ and the prediction with intercept = 0 (red line) is *674.1*, whereas when the intercept is

estimated this sum is (blue line) *478.7*. Note also the difference in explanatory power in the two panels. During deglaciation the blue regression line explains three fourths of the variance in $\Delta T$ whereas during glaciation less than half of the variance is explained. The interpretation is that during glaciation factors other than $CO_2$ account for the variation in $\Delta T$.

A more recent study (Snyder 2019) presents a Pleistocene data set that contains 799 records, evenly spaced in steps of

1000yr. Modeling, subjective probabilities and Monte Carlo analysis are applied in Snyder (2019) to quantify uncertainties in data and variable relationships. Changes in temperature with respect to the average over the last 5000 years are regressed on the total change in forcing $(\Delta R)$ from Green House Gases, Land Ice, dust and vegetation. A good comparison with Martinez-Boti et al (2015) is obtained by considering just the medians of the various terms. With respect to Snyder (2019) data, regressing the median of $\Delta T$ (termed Global Annual Surface Temperature GAST) on the median of $\Delta FGHG$, with

intercept, yields:

$$\Delta T(\Delta FGHG) = 1.962 \times \Delta GHG - 0.788 + e; \; \Delta T(556) = +6.49K$$

which compares well with eqn. (3), above, especially if we consider that Snyder (2019) estimates change in temperature with

respect to the average surface temperature over the last 5000 years which, according to Martinez-Boti et al (2015), is 0.45K higher than the pre-industrial reference value in their analysis. Hence we should compare 6.49 + 0.45 = +6.94K to +6.85K from eqn. (3). The medians of the separate forcing terms $\Delta R_{[GHG]}$, $\Delta R_{[LI]}$, $\Delta R_{[AE]}$, $\Delta R_{[VG]}$ are all highly correlated with each other and with $\Delta T$. Separating these forcing contributions means assuring that, on 1000yr timescales, land ice, dust and vegetation forcings are not already bleeding into the GHG forcing. This issue is avoided by focusing just on GHG forcing.

The penalty is that conclusions are wedded to the 1000yr timescale.

Summary statistics comparing results with and without *B = 0* are presented in Table 1: the partitions Pre and Post *424 kaBP* are intersected with deglaciation and glaciation periods.



### Summary and discussion

Extending previous analyses of Pleistocene climate data of Martinez-Boti et al (2015) and Snyder (2019), one key finding is that imposing a zero intercept tends to mask differences between the climate sensitivity parameter in different subsets of the Pleistocene data. This assumption produces inferior predictions relative to regressions in which estimates of intercept are exploited. Ultimately, this prevents us from appreciating the role of explanatory power in comparing the climate sensitivity

parameter in different physical situations.

Explaining why the climate sensitivity parameter is higher during deglaciation than during glaciation is a challenge which deserves attention. One obvious feature is the fact that deglaciation generally transpires faster than glaciation. It may be that certain negative feedbacks with intermediate time scales are allowed to play out during glaciation but are less effective

during rapid deglaciation. For example, carbon fertilization would draw down atmospheric $CO_2$. During glaciation the retreat of plants would tend to retard the cooling and slow the glaciation process. During deglaciation the advance of plants would tend to retard the warming process. However, if the retreat of land and sea ice happened quickly, other changes such as growth of the Hadley cells and desertification might overwhelm the advance of plants and disable this negative feedback. Oceanic $CO_2$ outgassing during the last deglaciation has been proposed (Shao et al 2019) as a potential disequilibrium

process influencing atmospheric $CO_2$ concentrations. Aerosols, and glacial aerosols specifically, can interact with clouds and influence radiative forcing; paleo-aerosol concentrations are likely to have varied with glaciation pathways. However, the lack of robust reconstructions of glacial aerosol forcing is a key source of uncertainty in paleo-based estimates of climate sensitivity (Friedrich, and Timmermann, 2020). As a further possibility, Xie (2020) posits that non-uniform regional uptake of heat by oceans militates against equilibration with radiative forcing, with spatio-temporal variations in ocean state and

currents affecting global climate sensitivity.

These are some conceptual examples of how the interplay of process-related feedbacks with different time scales might alter climate sensitivity in different physical situations. The supplementary material shows that the climate sensitivity parameter is higher after *424KaBP*, than before. Such effects have been attributed to changes in orbital forcing, though a detailed

understanding of why heightened forcing raises climate sensitivity has not, to our knowledge, been found.

If the Earth's surface temperature response to $CO_2$ forcing on millennial time scales does indeed depend on the physical circumstances at the time of the forcing, predictions for the Earth's future response to heightened $CO_2$ forcing require knowledge of these complex non-linear interactions, understanding how they will evolve and on what timescales. Scrutiny

and clarification of assumptions regarding the intercept issue and further analytical investigation with stochastic uncertainty techniques should provide additional insights into the properties of the climate sensitivity parameter and inform discussion of contingent implications.



**Acknowledgments** The authors are grateful to Gavin Foster and Tom Gernon for subject matter advice, and to Gordon Woo for commenting on a draft manuscript.

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

230

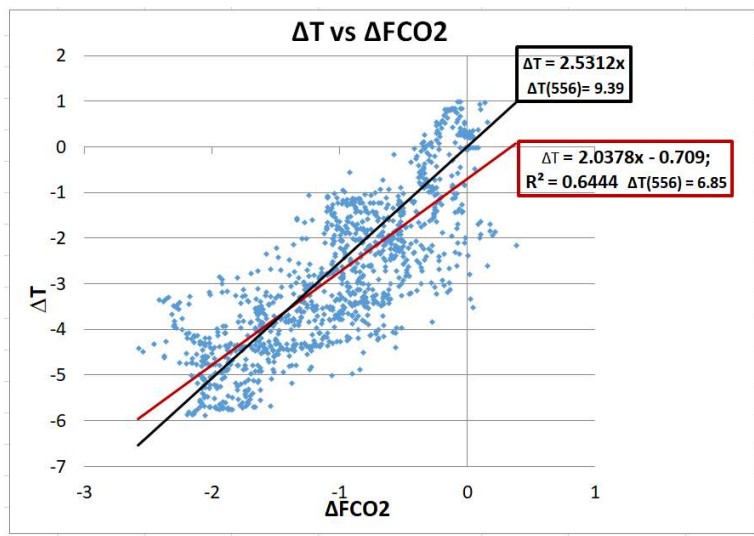

**Figure 1: Regression of ΔT on ΔFCO₂ for all Pleistocene data; with (red) and without (black) intercept. Out of sample values for doubling pre-industrial CO₂, ΔT(556) are also shown.**

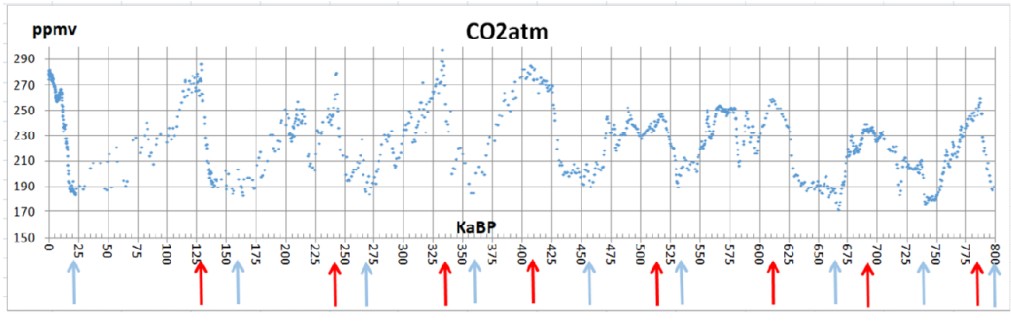





**Figure 2: CO2 concentrations as function of kaBP. Blue arrows indicate local minima, red arrows indicate local maxima.**

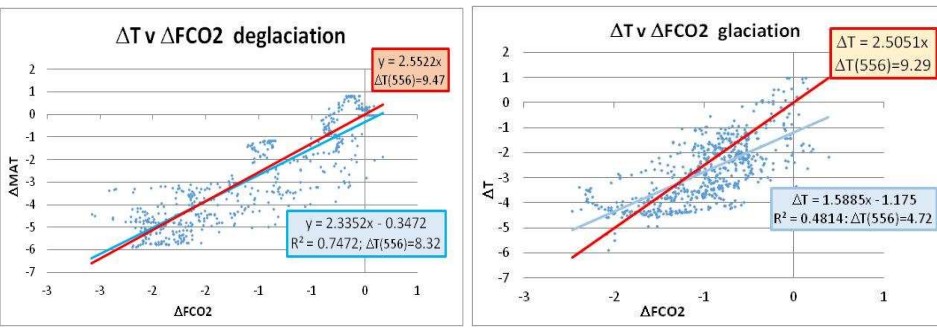

**Figure 3: Deglaciation, (left, increasing) and glaciation (right, decreasing) CO2 paths. Regression for B=0 stipulated in Red, with B estimated in Blue.**

|  | Intercept estimated | | | | | Intercept stipulated = 0 | | |
|---|---|---|---|---|---|---|---|---|
|  | $R^2$ | S | B | SE | ΔT(556) | S | SE | ΔT(556) |
| All (1096) | 0.644 | 2.038 | -0.709 | 0.978 | 6.851 | 2.531 | 1.047 | 9.391 |
| ALL Pre 424 (546) | 0.419 | 1.427 | -1.535 | 0.883 | 3.760 | 2.489 | 1.082 | 9.232 |
| All Post 424 (550) | 0.726 | 2.320 | -0.395 | 0.994 | 8.212 | 2.595 | 1.006 | 9.626 |
| De-Glaciation (increasing $CO_2$) | | | | | | | | |
| All (513) | 0.747 | 2.333 | -0.348 | 0.996 | 8.309 | 2.552 | 1.014 | 9.469 |
| Pre 424 (220) | 0.457 | 1.804 | -0.959 | 1.009 | 5.733 | 2.420 | 1.145 | 8.977 |
| Post 424 (293) | 0.830 | 2.631 | -0.215 | 0.916 | 9.545 | 2.798 | 0.781 | 10.381 |
| Glaciation (decreasing $CO_2$) | | | | | | | | |
| All (583) | 0.481 | 1.588 | -1.175 | 0.908 | 4.718 | 2.505 | 1.076 | 9.294 |
| Pre 424 (326) | 0.352 | 1.170 | -1.833 | 0.759 | 2.507 | 2.593 | 1.061 | 9.621 |
| post 424 (257) | 0.538 | 1.744 | -0.782 | 0.980 | 5.689 | 2.359 | 1.080 | 8.752 |
| Low, Medium, High $CO_2$ | | | | | | | | |
| <210 (305) | 0.022 | 0.485 | -3.576 | 0.779 | -1.778 | 2.367 | 0.902 | 8.781 |
| 210…250 (493) | 0.195 | 1.913 | -0.943 | 0.953 | 6.156 | 2.810 | 0.978 | 10.424 |
| >250 (298) | 0.201 | 2.711 | -0.396 | 1.096 | 9.664 | 3.670 | 1.123 | 13.614 |

**Table 1: Summary of data subset regressions. $R^2$ is the fraction of variance of ΔT explained by the regression, S is the climate sensitivity parameter, B is the intercept, SE is the standard deviation of the difference between predictions and true values, and ΔT(556) is the projected ΔT corresponding to a doubling of the pre-industrial CO2 concentration (278 ppmv). With the exception**



of "All" and "Pre 424" all S values are outside the 90% confidence bands of the other values within each group. The number of samples is in parentheses.