# Peer review of "Technical note: Pleistocene climate sensitivity to CO2 forcing is path dependent in reconstructions"

_Climate of the Past, 2020_

## Referee Comment (RC1) · Anonymous Referee #1 · 20 May 2020

This paper is about linear regressions in paleo data between global mean temperature change $\Delta$T and radiative forcing of $CO_2$ (called $\Delta$F $CO_2$), from which, in principle the slope of the regression ($\Delta$T / $\Delta$F $CO_2$) might be used as a paleo-data based estimate of climate sensitivity $S$.

The paper is set up as "technical note" which should, according to the guides to authors "report new developments, significant advances, and novel aspects of experimental and theoretical methods and techniques which are relevant for scientific investigations within the journal scope".

Actually, I can not see any of such things in this draft. One might argue, that the authors

make a case, that such linear regressions gives different response if the regression lines are forced through the origin, as suggested by some authors. Here, the authors try to argue for those regressions which allow an offset / bias in y, while others have argued, that maybe a non-linear regression might therefore be necessary to account for a state-dependency (von der Heydt et al., 2016; Köhler et al., 2017). Furthermore, they argue that regressions differ for glaciations or deglaciations (what they call partitioning by path). This is nothing new, and has been analysed in Figure 3 in Köhler et al. (2017), although for the relation of ΔT and ΔF caused by $CO_2$ and land ice, but see below on the difference and how useful it is to make the distinction (as done by the authors) for only ΔT and ΔF $CO_2$.

Being a technical paper, I do not even see progress in the regression analysis. The two data sets analysed do not consider the uncertainties in the individual data points, which are known to influence the regression analysis, and the resulting $r^2$ largely, e.g. see Press et al. (1992). Though, these uncertainty in both x and y has been included in other studies (e.g. Köhler et al., 2017; Snyder, 2019) (so, again no progress).

Furthermore, if paleo data are used to estimate for climate sensitivity from it, which is a measure typically used for an estimate of global warming as response to a doubling of $CO_2$ it is absolutely necessary to account for slow feedbacks in the climate system, mainly the land ice albedo feedback (PALAEOSENS-Project Members, 2012). If not accounted for the resulting $S$ will be much too high and is completely useless for any application on future climate change (e.g. for $2\times CO_2$). For example, the number for $S$ only caused by $CO_2$, called $S_{[CO_2]}$, based on data of the last 800 kyr was 3.1 K/(W/m$^2$), which is reduced to 1.1 K/(W/m$^2$) if land ice is considered (called $S_{[CO_2,LI]}$, and down to 0.7 K/(W/m$^2$) if all the available greenhouse gases and slow feedbacks are included, then called $S_{[GHG,LI,AE,VG]}$, see Table 2 in PALAEOSENS-Project Members (2012).

Thus, the given "explanation" of 64% of temperature change by $CO_2$ radiative forcing is only stating a statistical relation, but no explanation at all, and is due to a lot of missing processes simply wrong.

This paper brings nothing new, and what it shows is in all aspects too short and too simplified. It therefore should be rejected without any further revision. Even a major effort can not bring it to a paper suitable for publication, not for a general research paper and certainly not in the category of a technical note.

**Minor issues:**

1. Martínez-Botí et al. (2015) is a paper on the Pliocene and temperature and $CO_2$ said to be taken from that paper are certainly also only taken from somewhere else. Thus, the original references are missing here, eg $CO_2$ is taken from the ice core community.

**References**

Köhler, P., Stap, L. S., von der Heydt, A. S., de Boer, B., van de Wal, R. S. W., and Bloch-Johnson, J.: A state-dependent quantification of climate sensitivity based on paleo data of the last 2.1 million years, Paleoceanography, 32, 1102–1114, doi:10.1002/2017PA003190, 2017.

Martínez-Botí, M. A., Foster, G. L., Chalk, T. B., Rohling, E. J., Sexton, P. F., Lunt, D. J., Pancost, R. D., Badger, M. P. S., and Schmidt, D. N.: Plio-Pleistocene climate sensitivity evaluated using high-resolution $CO_2$ records, Nature, 518, 49–54, doi:10.1038/nature14145, 2015.

PALAEOSENS-Project Members: Making sense of palaeoclimate sensitivity, Nature, 491, 683–691, doi:10.1038/nature11574, 2012.

Press, W. H., Teukolsky, S. A., Vetterling, W. T., and Flannery, B. P.: Numerical recipes in Fortran, second edition, Cambridge University Press, Cambridge, 1992.

Snyder, C. W.: Revised estimates of paleoclimate sensitivity over the past 800,000Âăyears, Climatic Change, 156, 121–138, doi:10.1007/s10584-019-02536-0, 2019.

von der Heydt, A. S., Dijkstra, H. A., van de Wal, R. S. W., Caballero, R., Crucifix, M., Foster, G. L., Huber, M., Köhler, P., Rohling, E., Valdes, P. J., Ashwin, P., Bathiany, S., Berends, T.,

van Bree, L., Ditlevsen, P., Ghil, M., Haywood, A., Katzav, J., Lohmann, G., Lohmann, J., Lucarini, V., Marzocchi, A., Pälike, H., Baroni, I. R., Simon, D., Sluijs, A., Stap, L. B., Tantet, A., Viebahn, J., and Ziegler, M.: Lessons on climate sensitivity from past climate changes, Current Climate Change Reports, 2, 148–158, doi:10.1007/s40641-016-0049-3, 2016.

---

## Author Comment (AC1) · 20 May 2020

Thank you for your comment. Please tell us and other scientists what, fundamentally, is wrong in your opinion, and please cite the references where similar findings were published.

---

## Author Comment (AC2) · 24 May 2020

**What's New, What's Wrong: Response to Reviewer RC1**
Roger M. Cooke and Willy Aspinall
May 24, 2020

Thank you for reading the paper and for your comments. The paper offers a simple perspective on analyzing paleo data, it is not concerned with the provenance of the data. This does not require a long exposition. We cited several authors pointing to "state dependency" in the paleo climate sensitivity parameter. Many authors suggest non-linear regression. Problems with some of these approaches motivated our study.

Since the reviewer finds nothing new, we call out the elements which are new, in our opinion.
(1) The first "new" aspect of our analysis -- i.e. new to the discussion of paleo climate sensitivity -- is to point out that partitioned linear regression is another way to explore state dependence with certain advantages, in our opinion. One well known feature of non linear regression is that the best fit can behave wildly out of sample. Indeed, a cubic fit to the data in our Figure 1 allows temperature increase for forcing below *-2.5$Wm^{-2}$* (both with and without constrained intercept).
(2) Another "new" facet draws attention to reasons for not constraining the intercept to equal zero. It may, and in this case does, happen that the regression line through the origin is a worse predictor of the dependent variable than simply predicting the mean of the dependent variable for all values of the independent variable. We have not found this insight in the cited literature.
(3) Nor have we encountered recognition in the literature that $R^2$ does not correspond to the fraction of explained variance if the intercept is constrained to zero. ("Fractional explained variance" is statistical parlance, meaning the fraction by which variance is reduced by the statistical model. In a physical sense, statistics explains nothing). I n this light, strong arguments are needed to force the intercept to zero. The arguments we found in the cited literature are weak. "*However, note that here a necessary condition for the calculation of $S_{[X]}$ over the whole range of $\Delta R_{[X]}$, but not for the analysis of any state dependency, is that any fitting function crosses the origin with $\Delta R_{[CO2,LI]} = 0Wm^{-2}$ and $\Delta Tg = 0K$, implying for the fitting parameters that a* [the intercept] *= 0. This is also in line with the general concept that without any change in the external forcing, no change in global mean temperature should appear.*" (Kohler et al 2015, p1808). Our Figure 1 shows only $CO_2$ forcing, but the following remarks also apply when land ice forcing is included ($\Delta F_{CO2LIVDW11}$ from Martinez-Boti et al 2015).
There is substantial noise in the data. Thus, focusing on $\Delta F_{CO2} \sim 0$, values of $\Delta T$ vary from *1K* to *-3.5K* (with $\Delta F_{CO2LIVDW11}$ this is *1K* to *-2K*). If we constrain the regression line to pass through the origin, then we must explain why the deflections at $\Delta F_{CO2} = 0$ strongly tend to drive $\Delta T$ down, while those at $\Delta F_{CO2} = -2$ strongly tend to drive $\Delta T$ up. The attempt to circumvent the intercept issue leads to questionable mathematics: "*For the calculation of mean values of $S_{[CO2,LI]}$, we then analyse the $S_{[CO2,LI]} - \Delta R_{[CO2,LI]}$ space in a second step, where $S_{[CO2,LI]} = \Delta T_g \times \Delta R^{-1}_{[CO2,LI]}$ is first calculated individually for every data point and then stacked for different background conditions (described by $\Delta R_{[CO2,LI]}$). In doing so, we circumvent the problem which appeared in the $\Delta Tg - \Delta R[X]$ space that the regression function needs to meet the origin. Some of the individual values of $S_{[CO2,LI]}$ are still unrealistically high or low; therefore, values in $S_{[CO2,LI]}$ out- side the plausible range of 0–3 $K W^{-1} m^2$ are rejected from further analysis.*" (Kohler et al 2015, p1808). Studying the dependence of random variables $\Delta T_g$ and $\Delta R_{[CO2,LI]}$ by studying the mean or distribution of their ratio $S_{[CO2,LI]}$ is problematic. Putting aside issues of stability and truncation, consider two independent uniform variables on *[-10,-1]*, called *T* and *R*. By definition there is no dependence, yet the mean of *T/R* is *1.41*. Suppose we examine the state dependence of *T/R* on *R*. The conditional mean $\mu_{T|R=r}$ of T, given *R = r*, is *-5.5*, independent of *r*. However, the ratio *[$\mu_{T|R=r}$ ] / r* increases from *0.55* to *5.5* as *r* goes from *-10* to *-1*. Statisticians estimate the coefficient of linear dependence of *T* on *R* as *COV(T,R) /VAR(R)*, which has dimension *[T]/[R]*.

Understanding causes of deflections from a trend line is important.

(4) We have not seen fractional explained variance used as a diagnostic in the cited literature. $\Delta FCO_2$ accounts for *64%* of the variance in $\Delta T$ over the full Pleistocene data set. During deglaciation it accounts for *75%* and during glaciation it accounts for *48%*. Does that tell us something?

(5) Moreover, before *424 KaBP*, $\Delta FCO_2$ accounts for *42%* of the variance of $\Delta T$ and after *424 KaBP*, *73%*. This is also not found in the cited literature.

(6) $\Delta FCO_2$ has low explanatory power on partitions into low, medium and high $CO_2$. Different physical situations with the same reconstructed forcing can have different global surface air temperatures. Perhaps these facts can help us understand those differences. (Parenthetically, we note that Martínez-Botí et al (2015) over-samples the recent past. Removing this feature did not materially affect our results, and similar results are obtained with the dataset of Snyder (2019), which used *1000y* time steps.)

Much of the literature emphasizes Land Ice forcing, and the fact that this must be removed for predicting the effects of doubling $CO_2$ when the land ice is vastly reduced. We looked at this and eventually decided not to use these forcing terms as predicting the future was not our goal. We take advantage of this opportunity to share the following:

In Martínez-Botí et al (2015), three versions of Land Ice forcing are considered: $\Delta F_{CO2LIVDW11}$, $\Delta F_{CO2LIR09E12}$ and $\Delta F_{CO2LIR14}$ which include $CO_2$ forcing (see Martínez-Botí et al 2015 for detailed definitions). When regressing $\Delta T$ on these Land Ice forcing terms, the climate sensitivity parameter is lower than regressing on $\Delta F_{CO2}$. At the same time, these forcings account for more of the variance in $\Delta T$. The lower values of $S$ are explained by the wider range of values of the land ice forcing terms. The strongest effect occurs with $\Delta F_{CO2LIVDW11}$. The linear regression coefficient of $\Delta T$ on $\Delta F$ is $COV(\Delta T, \Delta F)/VAR(\Delta F)$. For $\Delta F = \Delta_{FCO2}$ these values are *0.85/0.42 = 2.04*. For $\Delta F = \Delta F_{CO2LIVDW11}$ they are *2.18/1.99 = 1.096*. Quadrupling the variance of the forcing term overwhelms the doubling of the covariance term, roughly speaking.

If we remove the $\Delta F_{CO2}$ and regress $\Delta T$ on $\Delta F_{CO2LIVDW11} - \Delta F_{CO2}$, something curious happens. $\Delta F_{CO2LIVDW11} - \Delta F_{CO2}$ yields a better predictor of $\Delta T$ ($\mathbf{R^2}$=0.94) than $\Delta F_{CO2LIVDW11}$ ($\mathbf{R^2}$=0.89). This suggests that $\Delta F_{CO2LIVDW11}$ may incorporate information on $\Delta T$ to the extent that $\Delta F_{CO2LIVDW11} - \Delta F_{CO2}$ becomes a proxy for $\Delta T$.

Finally, we believe that one of the fruits of the arduous work that has gone into preparing these high value paleo climate data sets is that others, from neighboring disciplines, can perhaps bring new ideas and tools to bear in analyzing these data. As non-specialists in paleo climate we have benefited enormously from the inclusive and supportive atmosphere within the paleo climate community, and look forward to strengthening these collaborations.

**References**

Köhler, P., de Boer, B., von der Heydt, A. S., Stap, L. B., and van de Wal, R. S. W. On the state dependency of equilibrium climate sensitivity during the last 5 million years, Climate of the Past 11(12):1801-1823, doi:10.5194/cp-11-1801-2015, (2015).

Martínez-Botí, M. A. et al. Plio-Pleistocene climate sensitivity evaluated using high-resolution $CO_2$ records. Nature 518, 49–54; doi:10.1038/nature14145 (2015).

Snyder, C.W. Revised estimates of paleoclimate sensitivity over the past 800,000 years. Climatic Change 156, 121-138; doi:10.1007/s10584-019-02536-0, (2019).

---

## Referee Comment (RC2) · Anonymous Referee #2 · 25 May 2020

In this study the authors set out to demonstrate that climate sensitivity was different during times of glaciation, than during times of deglaciation. They do so on the basis of a number of statements and assumptions, which unfortunately reveal a considerable lack of study of the relevant literature, and they arrive at a result that is not tested for robustness because input uncertainties are entirely ignored. Moreover, the interpretation of results is described in vague and speculative terms, without any testing, and without evaluation of the extent to which the inferences are a result of the uncertainties (because these were ignored). The writing is assuming a lot of pre-existing knowledge and understanding of the topic, or a major amount of reading of literature (including literature not referred to): the paper definitely is not a stand-along effort, which I don't

understand given that the journal is not imposing space limitations. Proper embedding in existing literature, introduction of concepts and definitions, and discussion of uncertainties and their propagation would be a must with this study.

But first of all: what is the study trying to establish? That glaciation and deglaciation pathways have different climate sensitivities? – Why is that important? The introduction doesn't tell us. – Also, is it novel? The introduction doesn't tell us. – Does it make a difference relative to other studies of state dependence? Effectively, this paper only diagnoses a difference, and doesn't really address why it might exist, whereas existing state-dependence work diagnosed the difference and at least evaluated why it might exist by considering the various forcings/feedbacks (among many others: Palaeosense, Nature 2012 with extensive detail in the Supplement; Köhler et al. Clim. Past, 2016; Köhler et al. Paleoceanography 2017; Köhler et al. Geophys. Res. Lett. 2018; von der Heydt et al., Geophys. Res. Lett. 2014; von der Heydt et al., Curr. Clim. Change Rep. 2016; von der Heydt and Ashwin, Dyn. Stat. Clim. Sys. 2016; Goodwin, 2018 Earth's Future, and many more. Some referred to in this study, and many not referred to at all). – Does it present more robust documentation of climate sensitivity state dependence? Well, it might have done, but it doesn't. First, the detection method here still uses an arbitrary subdivision of glacial cycles, albeit not between glacials and interglacials, but between "glaciation state" and deglaciation state", but it does so without giving a firm rationale why that might give a better view of state dependence. Second, the cut-off points between states are rather arbitrarily chosen: local minima and local maxima in $CO_2$... Glaciation state, does that not refer to glaciation (i.e. ice-volume change) anymore? Since when is $CO_2$ a measure for glaciation state? Sure, it may be a proxy for glaciation state, but why not go for a measure of the actual climate state change, rather than one of the forcings/feedbacks? Or at least evaluate if that gives a different answer or not... Moreover, the choice of "local" extremes is not explained. Local in what sense? Over what sort of timescales? And why are extremes a good choice, given that they may be extra sensitive to outliers? What happens if glaciation state is determined on the basis of first time derivatives?

Some smoothing/timescale of consideration will need to be decided for that also, and that would also need to be argued. None of that is even discussed here. We're simply confronted with hand-picked boundaries to allegedly different glaciation states. This opens the analysis up to major bias from subjective choices, and I suspect that that might be one of the problems that determines different sensitivity before and after the MBT at ∼425 ka. Or at least that it might be at the heart of that change – it's something that should have been investigated. Third, no account is given AT ALL of uncertainties. Both in terms of forcing, temperature, and age models. Nothing... How is that a complete analysis? What are the uncertainties involved, how do the uncertainties propagate, and – importantly – why would ordinary least squares analysis be appropriate, if there are uncertainties in X and Y? From mathematicians, I would expect extensive exploration of this issue, and a complete presentation of Total Least Squares analyses as well. And then all the uncertainties propagated, both in the analyses, and in the T(556) projections. And then an analysis of whether the different values found are really different, and what the statistics of those comparisons are. But there is nothing like it. ...

Overall, this is a paper addressing a problem that is rather well investigated, without referring to key studies. It is doing analyses that might compare favourably with those in the literature (although I would argue that they don't), but which are substantially below par for mathematicians who do nothing more than diagnosing some values (i.e., there is no real effort to come to a sound explanation, apart from some rather vague waffle in a paragraph at the end).

And to put a final nail in it: the Palaeosens study worked out a mathematical framework for reporting S. In this study, there is no reference at all to that, and this means that we're back to the 1990s in terms of definitions. That is, there is no real definition used, and where a choice is made (the study works in S[CO2] space), it is not argued why that's an appropriate one for the context studied. In fact, on palaeo timescales, most if not all slow feedbacks behave sort of like forcings. But especially the carbon cycle

feedback will have fast components too, which means that they behave like feedbacks (See Rohling et al., ARMS 2018). We cannot distinguish the proportionality between these types of behaviour, and palaeostudies have no choice but to consider all carbon cycle changes as slow feedbacks, which are then dealt with as forcings when working out the climate sensitivity parameters. Yet working in terms of S[CO2] only, vastly different T-change values will be found through time because of different influences of the slow ice-albedo (and even vegetation-albedo) changes (Palaeosens 2012). There are issues in evaluating slow feedbacks over relatively fast (millennial-scale) events, because the system may on those timescales not reach full equilibrium T response. Rohling et al (2018) demonstrated this in hypothetical scenarios, and suggested that elevated values of palaeoclimate sensitivity might be found in such cases. Interestingly, this study finds elevate sensitivity during deglaciations (which include a disproportionate amount of time covered by millennial-scale "see-saw" climate events), but there is just no discussion at all to assess whether this is a signature of the nature that was predicted. It all comes down to working in terms of CO2 forcing only, which for palaeo scenarios is not appropriate; certainly not if comparisons are to be made to modern climate change, which this study does with T(556). This is because the mathematical equivalence of palaeo S with modern or actuo S requires that all slow feedbacks are treated as forcings (Palaeosens, 2012). I find it an even worse affront to the major effort on climate sensitivity that has gone before this study that the classical palaeoclimate sensitivity studies of Jim Hansen are omitted entirely, while even Charney's classical work on climate sensitivity goes uncited. More specific comments: Line 16: "The climate sensitivity parameter S is somewhat loosely defined in the literature (Myhre et al 2013)." Yeah, well, maybe read a bit more around the subject before copying such a statement. Look at Palaeosens, and then ensure that your work is at least at some stage comparable with the definitions in there. Those are not loosely defined.

Lines 23-24: "The lower is $R^2$ the more of the variation in Delta-T must be attributed to factors other than Delta-FCO2." Maybe, but high $R^2$ can also be attributed to processes that covary closely with CO2, which is commonly the case in palaeo-records

because everything co-evolves. And nobody knows how the causal relationships run, except that there are tight feedbacks between all of the parameters. So it's wrong to assume that a high $R^2$ is proof of attribution to Delta-FCO2, or in other words it's wrong to think that CO2 is the forcing here... all slow feedbacks are equally important as forcings, and the impact of land-ice albedo change cannot be attributed to Delta-FCO2 (one can also argue it the other way around). Finally, orbital forcing – what process does it kick off? Do we know that? Or is all we know that it kicks of multiple tightly interwoven feedbacks, and that all slow feedbacks together determine the effective forcing to be considered in paleoclimate sensitivity assessment? I think the total avoidance of that discussion, and the total overlooking of other slow feedback "forcings" belies a highly simplified view of the world, which sort of invalidates the core of the assessment made here.

Line 46: Here the "climate sensitivity parameter" is dropped on the reader in line 46, but its definition is not given, and neither is its relationship to other climate sensitivity definitions. The text is merely a statement stack about climate sensitivity work, but does not provide any coherence about which definitions were used, how things compared, and so on... This is, overall, not an introduction to the problem, but merely a random statement collection that ignores a great wealth of research, and which does not work out the progression made through the various studies cited. How was the discussion advanced? What roadblocks were encountered and how were they overcome, or how does the present paper overcome them?

Line 39 and others "the dataset". Dataset for what? T? CO2? Ice Volume? All? It would be nice to get some more specific writing included throughout the paper, so the reader isn't left guessing so often. Another example would be "observational stidues" in line 43. Of what? modern? historical? palaeo? Or in Line 44, "combined with observational and modeling CMIP5 constraints" – what does this mean, that CMIP5 constraints have been modelled? Or that constraints from CMIP5 models are used? Grammar and precision of writing matter for ensuring that the meaning of arguments

comes across clearly. Or line 50: "non-linearity of CO2 forcing is said to depend on the CO2 data." What does that mean? That the data are not good, so that there is a non-linear artefact? Or that different CO2 records give different non linearities? How would that work, given that all CO2 records would be transformed into radiative forcing with the same logarithmic function? Or lines 51-52: "The need to distinguish actuo- and paleoclimate sensitivity over different time scales is emphasized (Rohling et al 2018)." Why drop that in here. This is part of the problem I highlighted before; the introduction is just a collection of statements, without (obvious) coherence. So what if that study said so. What does it mean for your study (I can think of some issues, as highlighted in my statements above). What do you do with it? Why is it important? And exactly the same questions apply again to the next random statement: "Averaged glacial and interglacial climate sensitivities are estimated (Shao et al 2019) using Earth system model simulations of the Last Glacial Cycle."

Line 68 "given the variations in S" – which variations in S? The ones you find in the following? If so, then isn't it a bit weird to here reveal a major implication of the study in the introduction already; in that case this sentence belongs in the discussion, no? Or perhaps this statement actually refers to previous work? In that case a reference seems needed.

A reference or references are needed for equation (1) – that is not your work.

In equation 3, S is not the "climate sensitivity parameter" as stated in line 89. Instead it is well-defined defined as S[CO2] in Palaeosens. It is the specific climate sensitivity parameter S[CO2], which refers to Earth System Sensitivity (not climate sensitivity). In Earth System Sensitivity, all T change is considered relative to only CO2 change. In climate sensitivity, the master term is that defined by Charney et al for modern climate, and specifically equilibrium climate sensitivity. To obtain the equivalent from palaeostudies, S[all slow feedbacks] is needed, which can be closely approximated by S[CO2,LI,VG], and still acceptably by S[CO2,LI] (e.g., Palaeosens 2012; Rohling et al 2018 ARMS). This goes to the heart of the present paper: it's not well defined, and

does not consider all forcings.

Line 95 and throughout. I am concerned about the level of precision reported (S and Delta_T to 3 decimal points), without any effort to discuss uncertainties and how these propagate into the answers - and how they affect the conclusion of different S for glaciating and deglaciating pathways; or whether the various uncertainties still allow sensible results from OLS regressions, or whether TLS regressions are needed.

Lines 111-112: "In this data, setting B = 0 inflates the S values and suppresses the differences over different partition elements." What partition elements are you referring to? The ones to be discussed in the next section? If so, then this sentence is out of place, and belongs in the next section. Or else the partitioning needs a bit of an intro here.

Line 116: simply write out "respectively." This is not a space-limited journal where abbreviations are needed.

Line 117: Why do you use CO2 to infer glaciation? Glaciation state refers to ice volume. By selecting it according to CO2 changes, you imply that CO2 is the cause or an immediate responder to glaciation state changes. This may be right (or right by approximation), but based on slow feedback processes it may just as well be wrong over timescales of a few thousand years. Moreover, you must assuming that the age difference between global T reconstructions and CO2 reconstructions is perfectly known. These are simplifications/ assumptions that need evaluation.

Line 120: "inception of MIS 11." This wording is confusing. Inception is commonly used for glacial inception. But here it is used for interglacial inception, which is commonly referred to as glacial termination. So, it may be technically correct, but still is confusing because of discipline-specific choices.

Lines 120-121: "the Earth's climate system changed around the inception of Marine Isotope Stage 11 in 424 kaBP, midway between a glacial maximum and a glacial mini-

mum." Changed in what sense? What happened? Any references? Hint: do a search on "Mid-Brunhes Transition."

Line 124: "the results show.." Which results? The ones in Figure 2 as called in line 127 (which should be Figure 3), or those in Table 1, which was the last thing you were discussing? As presented, this is not clear.

Line 125: now we're even in 4 decimal point precision... Do the math on uncertainties, and propagate them to your answers. Then report in sensible terms w.r.t. how robust the results are.

Line 126. Why T(522) here, when you were using T(556) everywhere else?

Line 127. I think Fig. 2 should be Fig. 3.

Line 131. Three quarters instead of three fourths?

Line 132. "The interpretation is that during glaciation factors other than CO2 account for the variation in Delta-T." Such as...? It all looks rather incomplete and ignorant of the very detail of the Palaeosens framework, and all the different "forcing" studies of Hansen, as well as Koehler et al (2010), Masson-Delmotte et al (2010), Rohling et al. (2012), and many others (several listed above in this review, and also to be found under authors such as van de Wal, von der Heydt, Stap, de Boer, and many many others (I don't feel called upon to do your literature search for you). These all indicate (and partially resolve) the "other factors" acting through time. Why not use that, and work up some tests of your findings? Why leave it hanging after a very simple regression exercise, and not push ahead toward a better understanding of the climate system? As is, what is this paper actually adding?

Line 134: "A more recent study." More recent than what?

Line 156. Actually, to really advance the debate, regressions should not use one independent and one dependent variable. We have uncertainties (considerable ones!) in both X and Y. These should be taken into account, and a Total Least Squares regression should be used. Uncertainties should be properly propagated, and answers should be considered within context of these uncertainties.

Lines 156-158: "Extending previous analyses of Pleistocene climate data of (Martinez-Boti et al 2015) and (Snyder 2019), one key finding is that imposing a zero intercept tends to mask differences between the climate sensitivity parameter in different subsets of the Pleistocene data." As written, this sentence strongly suggests that the previous studies imposed a zero intercept. That is a misrepresentation.

Lines 165-167: "During glaciation the retreat of plants would tend to retard the cooling and slow the glaciation process. During deglaciation the advance of plants would tend to retard the warming process." This is sloppy writing. Retreat and advance in what sense? Poleward? In terms of altitude? Both? And these vegetation changes need to be considered separately in the Palaeosense framework. And they do not respond only to carbon fertilisation, but mainly to T and humidity. But more importantly, this sentence represents mere unsubstantiated guesswork. It, in a primitive manner, reinvents the reasons why we need to correct for "forcings" due to the slow carbon-cycle and land-ice albedo, and vegetation feedbacks. If the authors would get more involved in palaeoclimate sensitivity definitions and previous studies, then they would have found ways by which to evaluate their supposition here.

Paragraph 162-175. This is just a summary of statements plucked from the literature, with little coherence or attempt to test the various proposed influences in the context of the analysis done in this manuscript. Work it out properly. Find out which effects are potential players, and by how much they could affect your result. The various influences that can be accounted for must be accounted for (see Koehler., 2016 for state dependence assessment under that approach; and see also Palaeosense and especially their Supplementary [notably Figs 4-6]).

Lines 179-180: "Such effects have been attributed to changes in orbital forcing, though a detailed understanding why heightened forcing raises climate sensitivity has not, to

our knowledge, been found." Have been attributed – by whom? References are in order here. And attributed how, and in what sense? Furthermore, in what sense was orbital forcing "heightened" Do you mean to say orbital insolation was stronger after 424 ka? Well, maybe show a record to demonstrate this then? Why does this manuscript stay so thin on the details all of the time? Why are the other slow feedbacks ignored, why is orbital forcing not shown or at least discussed? This is not a space-limited journal. Show us what you intend to say.

The very final paragraph is just jargon-filled waffle; it says very little in specific terms. Maybe the authors have an idea of what they mean to say here, but they certainly don't manage to communicate it in terms of an executable pathway for further research. In addition, I would again argue that the authors should first do some proper statistics, using uncertainties in X and Y, and propagating uncertainties from all input parameters to see if their perceived differences are statistically robust or not. They should also do a proper search of the literature, and express their work in a proper context of definitions of terms as has been laid out for palaeoclimate sensitivity - notably in Palaeosens. They also need to address the general ignoring of knowledge in the literature about the need to account for different slow feedbacks (which in palaeoclimate context present themselves as "forcings"). This is important because, when the authors look at only $CO_2$ changes, they are not comparing a consistent framework because of different processes and their different timescales. As a result, times of change over different timespans (glaciation v deglaciation) may be somewhat expected to show different responses. All the data exist for the authors to do the diagnosis as they do it here (but then properly, accounting for the uncertainnties), and to then test whether it's the omission of the various other slow feedbacks that explains the differences.

This paper needs a complete overhaul. As is, it does nothing for the state of understanding of the subject, and the presentation is full of holes. I recommend rejection.

---

## Author Comment (AC3) · 4 Jul 2020

Reply to Referee 2
Roger Cooke, Willy Aspinall
July 3 2020

We thank the referee for the consideration given to our article. Both referees are highly critical, yet neither renders an opinion on the fundamental methodological issue we raise: the presumption, in many existing regression analyses of Pleistocene climate data, that the intercept of the regression line must pass through the origin. For the published dataset of Martínez-Botí et al (2015), which we analysed, that assumption would be rejected in a standard $t$-test at the significance level $1.28 \times 10^{-34}$. Only when the assumption of a tie-in to the origin is discarded does an important feature of the data emerge: the rate of change in mean annual surface temperature with respect to $CO_2$ forcing is different during episodes of decreasing $CO_2$ (which we termed "glaciation") and increasing $CO_2$ (termed "deglaciation"). The differences are large and the explanatory power of the regressions in these episodes also show large differences. Of course, as mentioned, these regressions capture only effects that play out within 1000 year timescales. Effects of slower processes show up in the noise. Loss of explanatory power during glaciation suggests that slower processes are more important than in the more rapid deglaciation episodes.

We were trained to first apply a simple analysis of data to see the molar features. It helps to catch big mistakes before adding assumptions and detail. We may have missed something in our readings, but nowhere in germane literature have we found the above issues addressed. Accordingly, we believe a brief Technical Note is the appropriate format for raising this finding and associated regression issues. Many referee comments effectively enjoin us to abandon the brief Technical Note format. Unless that format is excluded we forego addressing these comments, except to say that the characterizations in lines 39 to 54 to which the referee takes umbrage are not our inventions but taken verbatim from the cited articles.

Neither referee has indicated where the following pertinent facts are identified in the literature, to wit:
(1) statistical evidence against forcing the intercept to zero,
(2) the differences in regression coefficients in de/glaciation episodes, and
(3) differences in explanatory power in de/glaciation episodes.

We further believe that the paleoclimate community would be well served by enlisting a referee who can render an opinion on the central methodological issues relating to regression analysis.

For good order, we mention that Snyder (2019) gives detailed analysis of various regression models, and partitions the data according to $\Delta T$, 450kaBP and temperature within 3.5C of the present. In Snyder's study, independent variables include forcings from $CO_2$, land ice, dust and vegetation -- reconstructed in various ways under various subjective assumptions. After spending considerable time reviewing those analyses, in our Technical Note to CP we decided, for reasons given in our reply to referee 1, to stick with the simple analysis, with $CO_2$ forcing as the sole independent variable. For convenience we reproduce here the considerations we articulated in our Technical Note:

"Much of the literature emphasizes Land Ice forcing, and the fact that this must be removed for predicting the effects of doubling $CO_2$ when the land ice is vastly reduced. We looked at this and eventually decided not to use these forcing terms as predicting the future was not our goal. We take advantage of this opportunity to share the following:

In Martínez-Botí et al (2015), three versions of Land Ice forcing are considered: $\Delta F_{CO2LIVDW11}$, $\Delta F_{CO2LIR09E12}$ and $\Delta F_{CO2LIR14}$ which include $CO_2$ forcing (see Martínez-Botí et al 2015 for detailed definitions). When regressing $\Delta T$ on these Land Ice forcing terms, the climate sensitivity parameter is lower than regressing on $\Delta F_{CO2}$. At the same time, these forcings explain more of the variance in $\Delta T$. The lower values of $S$ are explained by the wider range of values of the land ice forcing terms. The strongest effect occurs with $\Delta F_{CO2LIVDW11}$. The linear regression coefficient of $\Delta T$ on $\Delta F$ is $COV(\Delta T, \Delta F)/VAR(\Delta F)$. For $\Delta F = \Delta_{FCO2}$ these values are *0.85/0.42 = 2.04*. For $\Delta F = \Delta F_{CO2LIVDW11}$ they are *2.18/1.99 = 1.096*. Quadrupling the variance of the forcing term overwhelms the doubling of the covariance term, roughly speaking.

If we remove the $\Delta F_{CO2}$ and regress $\Delta T$ on $\Delta F_{CO2LIVDW11} - \Delta F_{CO2}$, something curious happens. $\Delta F_{CO2LIVDW11} - \Delta F_{CO2}$ yields a better predictor of $\Delta T$ *($R^2$=0.94)* than $\Delta F_{CO2LIVDW11}$ *($R^2$=0.89)*. This suggests that $\Delta F_{CO2LIVDW11}$ may incorporate information on $\Delta T$ to the extent that $\Delta F_{CO2LIVDW11} - \Delta F_{CO2}$ becomes a proxy for $\Delta T$."